# Meta-in-context learning in large language models

**Julian Coda-Forno**[1,2,*]   **Marcel Binz**[1]   **Zeynep Akata**[2]
**Matthew Botvinick**[3]   **Jane X. Wang**[3]   **Eric Schulz**[1]
[1]Max Planck Institute for Biological Cybernetics, [2]University of Tübingen - Tübingen, Germany;
[3]Google DeepMind - London, United-Kingdom
*{julian.coda-forno@tuebingen.mpg.de}

## Abstract

Large language models have shown tremendous performance in a variety of tasks. In-context learning – the ability to improve at a task after being provided with a number of demonstrations – is seen as one of the main contributors to their success. In the present paper, we demonstrate that the in-context learning abilities of large language models can be recursively improved via in-context learning itself. We coin this phenomenon *meta-in-context learning*. Looking at two idealized domains, a one-dimensional regression task and a two-armed bandit task, we show that meta-in-context learning adaptively reshapes a large language model's priors over expected tasks. Furthermore, we find that meta-in-context learning modifies the in-context learning strategies of such models. Finally, we broaden the scope of our investigation to encompass two diverse benchmarks: one focusing on real-world regression problems and the other encompassing multiple NLP tasks. In both cases, we observe competitive performance comparable to that of traditional learning algorithms. Taken together, our work improves our understanding of in-context learning and paves the way toward adapting large language models to the environment they are applied purely through meta-in-context learning rather than traditional finetuning.

## 1 Introduction

Large language models (LLMs) are taking not only machine learning research but also society by storm [1, 2, 3]. Part of what makes these models so persuasive is that their abilities reach far beyond what we expected pure language models to do. They can, among other things, solve challenging reasoning problems, including university-level math questions [4] or analogical reasoning tasks [5], out-of-the-box and without additional training.

Much of this power comes from what is known as in-context learning [6]. In-context learning (sometimes also called few-shot learning or few-shot prompting) refers to the ability of an LLM to improve at a given task after being provided with a number of task-relevant demonstrations. This ability sets LLMs apart from traditional models and led to a totally new paradigm – one which eschews finetuning of weights on task-specific data altogether and instead relies entirely on contextual information.

In the present paper, we ask whether the learning algorithm implemented through in-context learning can be improved through in-context learning itself (without the need for any further finetuning of parameters). To study this question, we conducted several experiments where we presented an LLM with multiple learning tasks in a sequence. In three distinct settings, we find evidence for the idea that the in-context learning abilities of an LLM can be recursively enhanced via in-context learning, thereby displaying a form of *meta-in-context learning*. Figure 1 provides a high-level overview of our approach on an example supervised learning task.

37th Conference on Neural Information Processing Systems (NeurIPS 2023).

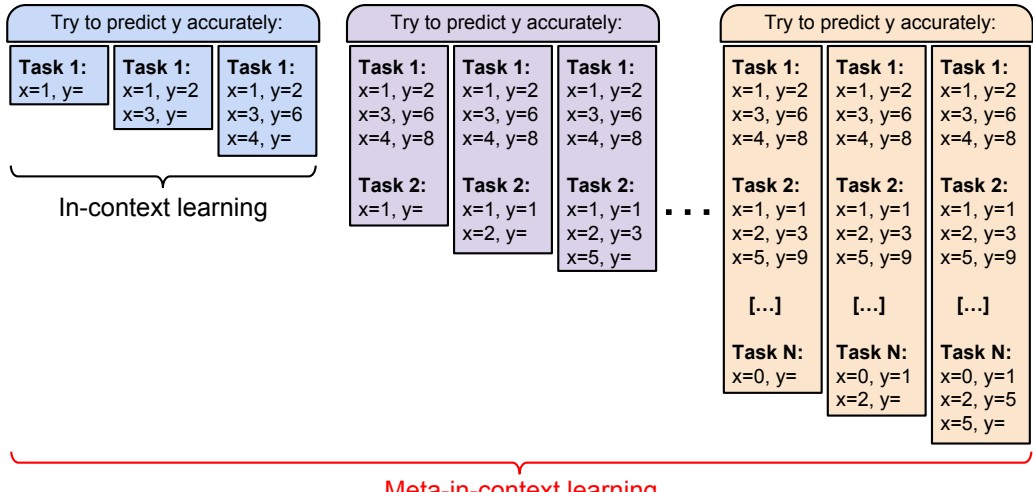

Figure 1: High-level overview of our approach on an example of multiple three-shot regression tasks. We present an LLM with $N$ learning tasks in a row. Improvement within a task indicates that the model is capable of in-context learning. If in-context learning improves across multiple learning tasks, the model is also capable of meta-in-context learning.

More specifically, we first investigate meta-in-context learning on two artificial domains: a supervised function learning task, and a two-armed bandit task. For both of them, we find that sequentially presenting LLMs with multiple learning problems boosts their in-context learning abilities. We then use these idealized domains to identify the drivers behind meta-in-context learning. We find that meta-in-context learning adaptively modifies priors over latent variables, ultimately leading to priors that closely resemble the true statistics of the environment. Furthermore, our analysis reveals that meta-in-context learning can not only be used to change prior expectations but is also capable of reshaping an LLM's learning strategies. Lastly, we apply our approach to two realistic domains. We first demonstrate that meta-in-context learning can be used to obtain a learning algorithm that is competitive with traditional algorithms on a benchmark of real-world regression problems. We then validated the transferability of our findings to an NLP benchmark, confirming the effectiveness of our approach in diverse contexts.

## 2 Related work

**In-context learning:** Recent work has shown that LLMs can improve their performance after being shown a few task-relevant demonstrations – an ability referred to as in-context learning [6]. When and why in-context learning emerges is a matter of ongoing debate, with different theories being proposed. Chan et al. [7] argued that properties of the training data – such as burstiness and non-stationarity – are key drivers behind in-context learning. Min et al. [8], on the other hand, found that ground truth demonstrations can be replaced with random labels while barely hurting performance, suggesting that the role of demonstrations is more to prime the model for a particular task. Finally, Xie et al. [9] suggested that LLMs internally need to infer latent variables to make better predictions about future word occurrences, thereby implementing a form of Bayesian inference.

**In-context learning can solve classical learning tasks:** If LLMs do apply some form of Bayesian inference, then one would expect an LLM to also be able to solve classical online learning tasks, such as regression or classification, purely through in-context learning. Previous research suggests that this is indeed the case. Lovre [10], for instance, tested GPT-3 on a range of low-dimensional classification and regression tasks and found that it was often on par with classical learning algorithms such as logistic regression. Likewise, Hegselmann et al. [11] tested an LLM's few-shot classification abilities on tabular data. They found that their approach outperforms prior deep-learning-based tabular classification methods on several benchmark datasets, and that performance further improves when

the model is provided with semantic information about the data.[1] LLMs are not only able to solve supervised learning problems but also simple reinforcement learning tasks. To provide one example, Binz & Schulz [12] evaluated GPT-3 on a two-armed bandit task and found that its performance exceeded that of human participants who did the corresponding psychological experiment.

**Meta-in-context versus classical meta-learning schemes:** Meta-in-context learning stands in contrast to classical meta-learning schemes in which one adapts a neural network to a distribution over learning problems by adjusting its weights [13, 14]. Historically, this approach has relied on recurrent networks [15, 16, 17, 18] but, more recently, researchers have also started to use transformer-based architectures [19, 20, 21]. For example, Garg et al. showed "transformers can be trained from scratch to perform in-context learning of linear functions" and that the resulting models achieve "performance comparable to the optimal least squares estimator." In a similar vein, von Oswald et al. argued that "transformers [can in principle] implement gradient descent in their forward pass" and provide empirical evidence that they indeed do so [19]. In contrast to these approaches, our approach adapts a model to a series of learning problems entirely through the context itself as opposed to updating weights.

## 3   Experimental setup

We used the public OpenAI Python API [22] to run all our simulations. This API provides access to several LLMs from the Generative Pre-trained Transformer (GPT) family. We ran all our simulations on the TEXT-DAVINCI-002 model, which is also known as GPT-3. We set the temperature parameter to zero (leading to deterministic responses) unless otherwise noted and retained the default values for all other parameters. It is important to note that all experiments performed in this paper rely entirely on the in-context learning abilities of an LLM, and do not involve any form of finetuning.

## 4   Learning one-dimensional functions

In our first set of experiments, we investigated GPT-3's ability for meta-in-context learning in a simple one-dimensional supervised regression task. In this setting, we provided GPT-3 with a list of input-target pairs from a given task and asked it to make accurate predictions for a new input value. This is one of the most fundamental machine learning problems, and its simplicity makes it an ideal testbed for an initial analysis.

### 4.1   Methods

> **Function learning prompt**
>
> You observe 5 machines that produce an output $y$ for a given input $x$.
> Each machine implements a different function.
>
> Machine 1:
>
> $[\dots]$
>
> Machine 5:
> $x = 52$, $y = -209$;
> $x = 18$, $y = -138$;
> $x = 60$, $y = [\text{insert}]$;

Every task $n$ consisted of $T = 5$ input-target pairs $(x_t, f(x_t))$, where $t \in [1, T]$ denotes the trial number. Each input-target pair was generated by a linear function of the form $f(x_t) = a^{(n)} x_t + b^{(n)} + \varepsilon_t$, where $a^{(n)}$ and $b^{(n)}$ are task-specific parameters drawn from a probability distribution $p(\mathcal{T})$. Inputs $x_t$ were sampled from $\mathcal{U}(0, 100)$ and the trial-specific additive noise $\varepsilon_t$ was sampled from $\mathcal{N}(0, 1)$. For our meta-in-context learning simulations, we considered prompts that include data from up to five tasks, each corresponding to a different underlying function. Following the schema

---

[1]However, it is important to note that their approach does not entirely rely on in-context learning, but involves some additional finetuning.

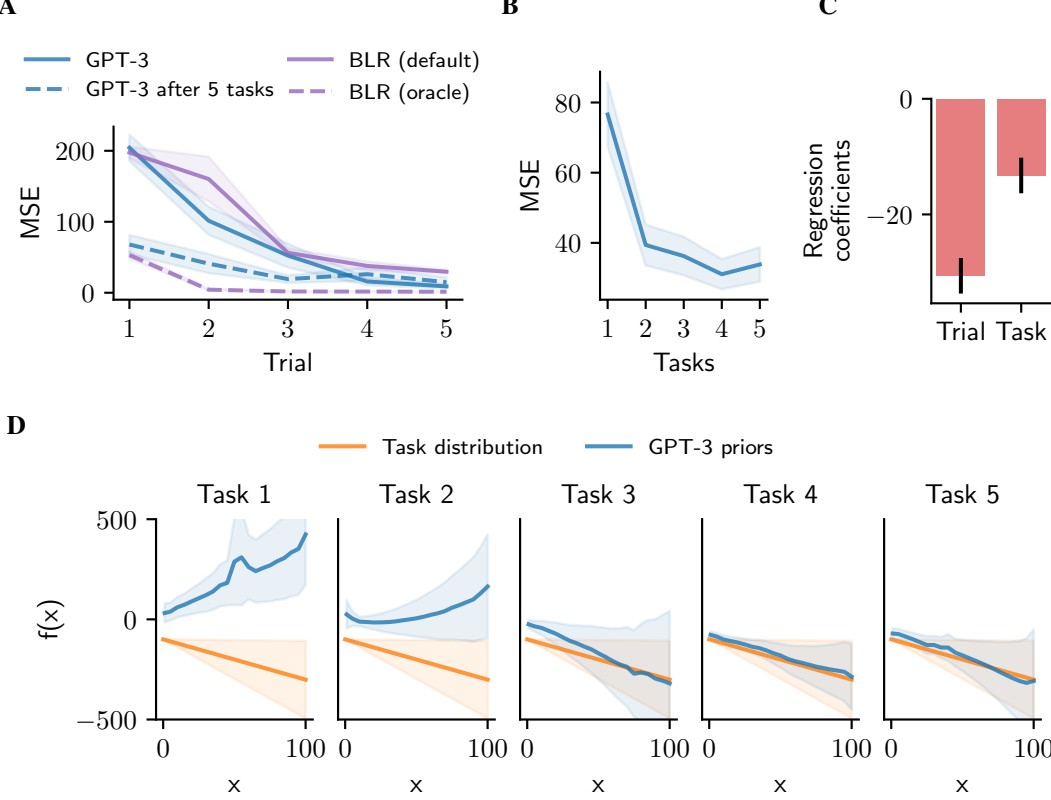

Figure 2: Meta-in-context learning on the one-dimensional regression task (100 simulations). Errors bars represent 95% confidence intervals. **A**: MSE across trials for different models. **B**: GPT-3's MSE averaged over trials for each task. **C**: Effects of trial and task for estimating the MSE. **D**: GPT-3's prior expectations across tasks (blue) compared to the true task distribution (orange).

outlined in Figure 1, we iteratively presented each data-point together with the history of previous observations, starting from the first trial in task one up to the last trial in task five. In the "function learning prompt" box above, you can see an example prompt that we provided to GPT-3.

### 4.2 Results

In preliminary simulations, we found that GPT-3 has a strong bias toward increasing positive functions. To demonstrate the potential of our approach, we wanted to investigate whether it is possible to overwrite this bias via meta-in-context learning. In order to achieve this goal, we sampled task-specific functions with a negative slope and intercept, i.e. $a^{(n)} \sim \mathcal{N}(-2, 1)$ and $b^{(n)} \sim \mathcal{N}(-100, 1)$, and evaluated how observing multiple such tasks influences the model's behavior.

**GPT-3 does in-context learning:** We first established that GPT-3 can learn within a single task without considering the effects of meta-in-context learning. To do so, we only examined the first task while ignoring all the subsequent ones. We found that performance as measured by the mean-squared error (MSE) improves with additional data-points as shown in Figure 2A (solid lines). GPT-3 matches (or even slightly outperforms) a Bayesian linear regression (BLR) model with default standard normal priors, indicating that in-context learning is able to solve the given problem. This is especially noteworthy since GPT-3 was never told that the underlying functions are linear, as opposed to BLR which has a built-in assumption of linearity.

**GPT-3 does meta-in-context learning:** Next, we investigated whether GPT-3 is capable of meta-in-context learning. For this, we inspected how performance changes across the five tasks. Figure 2A demonstrates that meta-in-context learning is beneficial by comparing performance between the first

and the final task (solid vs dashed lines). For reference, we also plotted the performance of a BLR model with access to the ground-truth data-generating distribution. Note that this model has access to privileged information and hence only serves as a performance upper-bound. Figure 2B shows a more detailed development of performance as we increase the number of tasks in the prompt.

To test whether the effects of in-context learning and meta-in-context learning are statistically meaningful, we fitted a linear regression model that included the trial and task number as independent variables on the MSE. We found statistically significant effects for both trial number ($\beta = -30.614 \pm 3.08$, $t = -19.514$, $p < 0.001$) and task number ($\beta = -13.26 \pm 3.08$, $t = -8.455$, $p < 0.001$), confirming that GPT-3 is capable of both in-context and meta-in-context learning (see Figure 2C).

**Meta-in-context learning is driven by adaptation of priors:** We speculated that GPT-3's performance improves during meta-in-context learning because it adapts its priors to true environmental statistics. In order to evaluate this hypothesis, we collected GPT-3's prior expectations before each task by asking it to make sequential predictions for 20 evenly spaced input values (using the same prompt template as above, but setting the temperature to one and feeding back the model's own predictions as the training data). We omitted outlier predictions that had absolute values of $10,000$ or larger for this analysis. The estimated priors indicate that GPT-3 has an initial bias toward increasing positive functions. However, already after two tasks it adapts its priors to decreasing negative functions, thereby closely matching the true environmental statistics (see Figure 2D). Furthermore, the prior for the intercept term is initially centered around zero but shifts toward the ground-truth value of $-100$ after three tasks.

Taken together, these simulations provide initial evidence that GPT-3 engages in meta-in-context learning, meaning that its in-context learning abilities can be recursively improved via in-context learning itself. We have suggested and empirically confirmed that GPT-3 accomplishes this by adapting its priors across multiple tasks.

**Meta-in-context learning is an emergent phenomenon:** In order to gain a deeper understanding of the phenomenon's characteristics, we pursued an examination into the progression of meta-in-context learning proficiency in relation to both model complexity and dataset size. To this end, we undertook an analysis encompassing smaller GPT-3 models, specifically text-ada, text-cabbage, and text-curie [22]. Our observations (which can be found in Figure 5A, Appendix A.1.1) reveal a noteworthy trend, wherein solely text-davinci-002 seems to exhibit meta-in-context learning capabilities for a negative linear function.

**Meta-in-context learning in open-source models:** To comprehensively explore meta-in-context learning, we extended our evaluation to encompass five distinct open-source models. This expansion allowed us to examine whether this phenomenon is also applicable to less opaque model architectures. Notably, while the majority of models (Falcon-40b [23] and Llama-2 [24] models) exhibited no indications of meta-in-context learning, intriguingly, MosaicML's mpt-30b [25] demonstrated an ability to manifest this phenomenon. We hypothesize that models trained in a regime with a longer context window are more suitable to show the emergence of this phenomenon. Indeed, mpt-30b has a context-size of 8000 tokens as opposed to the LLaMa-2 models which have been trained with a context window of 4096 tokens. This is speculative and we leave the analysis of what factors influence the emergence of meta-in-context learning as a future research question. The results are included in Figure 5B, Appendix A.1.1.

**Meta-in-context learning with non-linear functions:** We also aimed to determine whether GPT-3 could also adapt its meta-in-context learning to non-linear functions. In Figure 6, Appendix A.1.2, we present an analysis of a quadratic function conducted within the same experimental framework. Our findings indicate that the model indeed exhibits the ability to meta-in-context learn non-linear functions.

# 5 Experiments on two-armed bandit tasks

In a next step, we wanted to investigate whether our results from the supervised setting also transfer to a reinforcement learning paradigm. This setting adds an additional layer of complexity because it requires the agent to learn from its own experiences instead of having access to ground-truth solutions of previous tasks. In addition, it allows us to investigate learning strategies and how they evolve during meta-in-context learning.

## 5.1 Methods

For our simulations, we considered a simple two-armed bandit task, in which an agent repeatedly interacts with two slot machines. In each trial, the agent can select one of two machines and is rewarded based on a probability distribution that is associated with that machine. The agent's objective is to maximize the total amount of acquired points. We used a cover story that involves a gambler visiting different casinos, which has been used in human experiments with similar tasks [26, 12], to generate our prompts:

> **Two-armed bandit task prompt**
>
> You are going to different casinos that own two slot machines.
> Choosing the same slot machine will not always give you the same points, but one slot machine is always better than the other. Within a casino, your goal is to choose the slot machine that will give you the most points over the course of 10 trials.
> Each casino owns a different pair of machine.
>
> You have received the following points when playing in casino 1:
>
> $$[\dots]$$
>
> You have received the following points when playing in casino 5:
> - Machine J delivered 4.2 points.
> - Machine F delivered -7.4 points.
>
> Q: We are now performing trial 3 in casino 5. Which machine do you choose between machine J and machine F?
> A: Machine [insert].

## 5.2 Results

For each task, we sampled independent mean rewards for each machine from $\mathcal{N}(0, \sqrt{64})$. The actually obtained reward is generated using the mean reward that corresponds to the chosen machine plus some additive Gaussian noise sampled from $\mathcal{N}(0, \sqrt{32})$. To stay consistent with the previous section, we considered prompts that include data from up to five different tasks for our meta-in-context learning simulations (each of these tasks consisted of ten trials). We used letters to indicate the different slot machines. For each task, we randomly sampled two different letters without replacement to cancel out biases towards certain letters.[2]

**GPT-3 does in-context learning:** We again first tested whether GPT-3 learns within a single task. Figure 3A confirms that this is indeed the case. Performance (measured in terms of regret) improves over the first four trials and plateaus afterward. However, when comparing GPT-3 to two baseline algorithms – a greedy policy and an upper confidence bound (UCB) algorithm – we observed that it lags in performance prior to meta-in-context learning.

**GPT-3 does meta-in-context learning:** Like in our previous experiment, we found that GPT-3 improves during meta-in-context learning: as it observes more tasks, it gets better at solving new two-armed bandit problems (see Figure 3B). GPT-3's performance matches that of a greedy policy after having interacted with only four tasks and lags only slightly behind that of the UCB algorithm.

We quantified the effects of in-context and meta-in-context learning in a statistical analysis. To do so, we fitted a linear regression model including the trial and task number as independent variables on the trial-wise regret. We found significant in-context learning ($\beta = -0.221 \pm 0.012, t = -35.828, p < 0.001$) and meta-in-context learning effects ($\beta = -0.031 \pm 0.012, t = -5.002, p < 0.001$), thereby reproducing our results from the previous section (see Figure 3C).

**Meta-in-context learning is driven by adaptation of priors:** Following our earlier analysis, we speculated that part of this performance boost arises because GPT-3 adapts its priors toward the statistics of the tasks that were encountered during meta-in-context learning. To verify this, we

---

[2]We left out the letters "I" and "U" as we have seen empirically that they induce biases. We also randomized the order of these letters in the questions at each trial to mitigate recency biases.

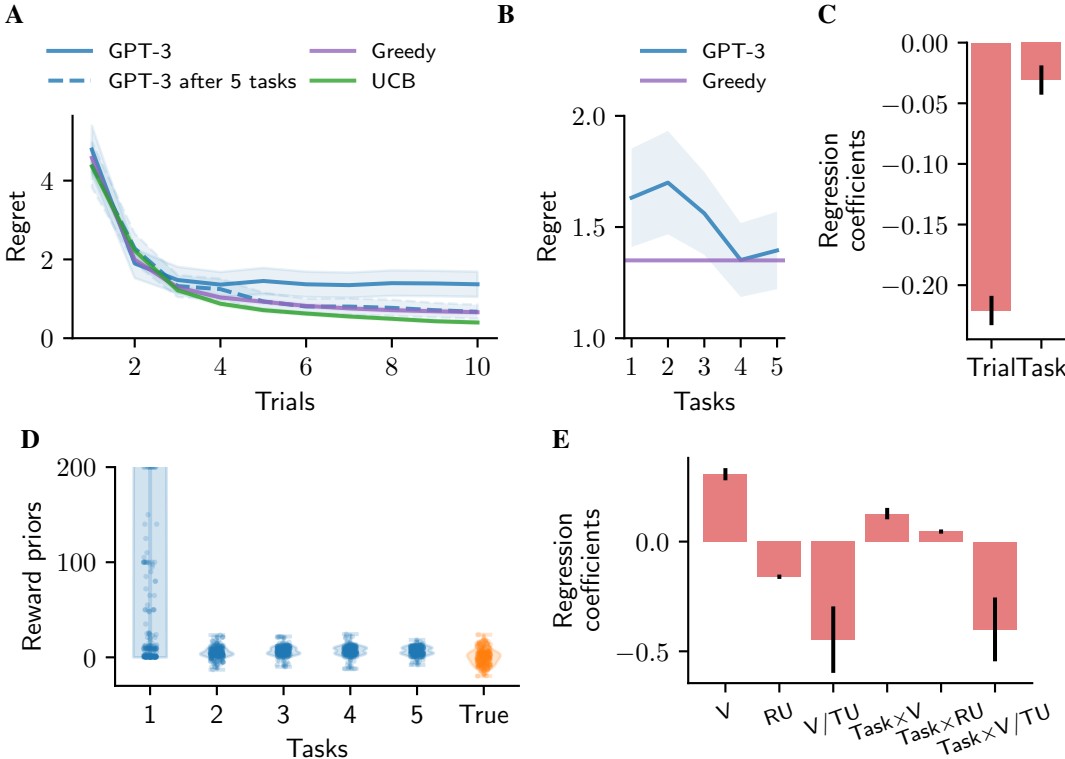

Figure 3: Meta-in-context learning on the two-armed bandit experiment (500 simulations). Errors bars represent 95% confidence intervals. **A**: Regrets across trials for different models. **B**: GPT-3's regrets averaged over trials for each task. **C**: Effects of trial and task for estimating the regret. **D**: GPT-3's prior expectation of rewards across games. **E**: Probit regression coefficients for different strategies and their interaction with task number.

probed its prior expectations before starting each task by asking "how rewarding do you expect machine X to be?" We set the temperature parameter $= 1$ and repeated this question five times to reflect a sampling from a prior distribution. Figure 3D visualizes the change of priors across tasks. Before the initial task, GPT-3 expects rewards to be distributed around larger positive values ($M = 545.22$, $SD = 6359$). However, at the end of meta-in-context learning, its priors match the true data-generating distribution closely ($M = 6.49$, $SD = 4.41$).

In addition to looking at the development of priors, the two-armed bandit setting also allows us to investigate whether any changes in strategies happen during meta-in-context learning. For this, we relied on an analysis originally proposed by Gershman [26]. The idea behind this analysis is to define a model that involves a parameterized combination of Boltzmann exploration, a UCB algorithm, and Thompson sampling, and then fit the parameters of this model to data generated by an agent (GPT-3 in our case). It is then possible to determine the extent to which the agent relied on a specific strategy by examining the resulting parameters.

We assume that the agent's beliefs over expected rewards at trial $t$ and action $a$ are captured by the normal distribution $p(r_{a,t}) = \mathcal{N}(\mu_{a,t}, \sigma_{a,t})^3$ and define the following probit regression model based on the parameters of these distributions:

$$p(a_t = 0|\mathbf{w}) = \mathbf{\Phi}\left(\mathbf{w}_1 V_t + \mathbf{w}_2 RU_t + \mathbf{w}_3 V_t/TU_t\right) \tag{1}$$

$$V_t = \mu_{0,t} - \mu_{1,t} \qquad RU_t = \sigma_{0,t} - \sigma_{1,t} \qquad TU_t = \sqrt{\sigma_{0,t}^2 + \sigma_{1,t}^2} \tag{2}$$

with $\mathbf{\Phi}$ denoting the cumulative distribution function of a standard normal distribution. Equation 1 recovers a Boltzmann-like exploration strategy for $[\mathbf{w}_1, \mathbf{w}_2, \mathbf{w}_3] = [c, 0, 0]$, a variant of the UCB

---

³Following Gershman [26], we obtained these distributions by running Kalman filtering equations on the previously observed data. More details on this analysis can be found in Appendix A.3.

algorithm for $[\mathbf{w}_1, \mathbf{w}_2, \mathbf{w}_3] = [c, d, 0]$, and Thompson sampling for $[\mathbf{w}_1, \mathbf{w}_2, \mathbf{w}_3] = [0, 0, 1]$ [27]. In our analysis, we furthermore included an interaction effect with task number for each factor to investigate how the applied strategies change during meta-in-context learning.

**Meta-in-context learning reshapes learning strategies:** We found a positive main effect of the value difference $V_t$ ($\beta = 0.307 \pm 0.027, z = 21.759, p < 0.001$), indicating that GPT-3 engages in Boltzmann exploration. GPT-3 becomes more greedy during meta-in-context learning as shown by the positive interaction effect between task number and $V_t$ ($\beta = 0.128 \pm 0.025, z = 9.633, p < 0.001$). Furthermore, we found negative main effects for both relative uncertainty $RU_t$ ($\beta = -0.160 \pm 0.010, z = -31.788, p < 0.001$) and the uncertainty-scaled value difference $V_t/TU_t$ ($\beta = -0.400 \pm 0.025, z = -5.393, p < 0.001$), suggesting that GPT-3 avoids uncertain options by default. However, we also found a slight, but significant, increase in UCB-based decisions during meta-in-context learning ($\beta = 0.046 \pm 0.010, z = 9.279, p < 0.001$). Figure 3E shows a visualization of all regression coefficients involved in this analysis.

The results presented in this section corroborate those obtained from the supervised setting. GPT-3 generally performed better in a two-armed bandit task after meta-in-context learning. We again observed that GPT-3 accomplishes this by adapting its priors across tasks. In addition, we investigated changes in strategies during meta-in-context learning and found that GPT-3 learned to perform better by exploiting more consistently.

# 6 Regression on real-world data

Next, we wanted to investigate whether our results obtained in the artificial domains scale up to real-world applications. To test this, we considered a multi-dimensional regression benchmark which consists of 60 different real-world datasets introduced in [28].

## 6.1 Methods

For each simulation, we randomly selected five different tasks from the benchmark and then sampled five points without replacement for each task. We used five features for all tasks. If a dataset contained less than five features, we omitted it from our analysis, which yielded 42 remaining datasets. For tasks exceeding five features, we used a sub-selection procedure that retained only the top five features based on their F-value with respect to the target variable evaluated in a univariate linear regression. To maintain a consistent regression loss across all tasks, we normalized both the feature and target spaces to the interval of $[-1, 1]$. The resulting prompts follow the general template outlined earlier:

> **Regression on real world data**
>
> You observe an input vector $x$ and have to predict the corresponding output $y$ as accurately as possible. You are given 5 different tasks.
>
> Task 1:
>
> $[\ldots]$
>
> Task 5:
> $x = [-0.81, -0.16, -0.78, -0.77, -0.45], y = -0.34;$
> $x = [-0.81, -0.63, -0.75, -0.83, -0.55], y = -0.68;$
> $x = [-0.97, -0.92, -0.97, -0.97, -0.82], y = [\text{insert}];$

### 6.1.1 Results

**GPT-3 does in-context and meta-in-context learning:** Following the previous experiments, we investigated the learning curves of GPT-3 and GPT-3 after meta-in-context learning. We additionally included two baselines in our analysis: BLR and a random forest model. We measured performance by the root-mean-squared error (RMSE). Figure 4A shows the learning curve for all these models. Like in the previous cases, we found that GPT-3 improves with additional data-points, confirming that it is capable of in-context learning in this setting. In addition, meta-in-context learning further improved performance. GPT-3 after meta-in-context learning matched the performance of BLR and was only slightly outperformed by the random forest model.

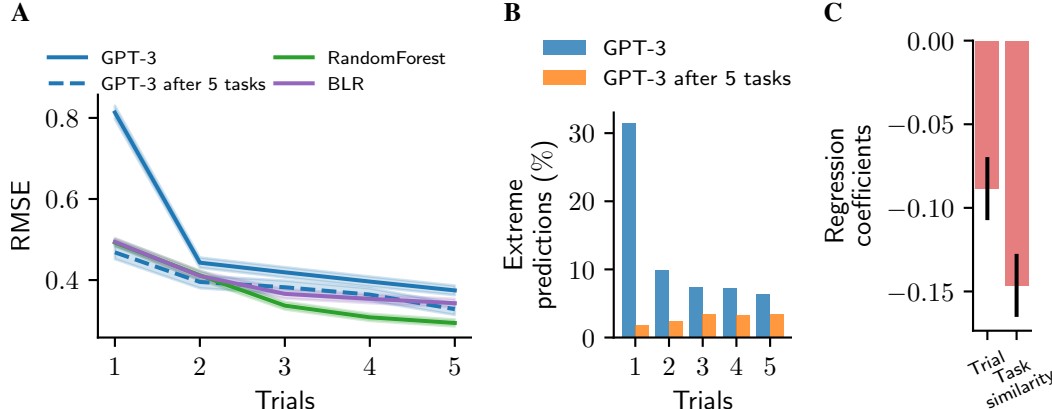

Figure 4: Meta-in-context learning on the real-world regression benchmark ($42 \cdot 50$ simulations). Errors bars represent $95\%$ confidence intervals. **A**: RMSE across trials for different models. **B**: Percentage of predictions outside or equal to the extremes of the squashed target range. **C**: Effects of trial and task similarities for estimating the RMSE.

**Meta-in-context learning constrains predictions:** Most of the improvement of meta-in-context learning seems to come from zero-shot performance. We hypothesized that this is the case because GPT-3 acquires an understanding of potential target value ranges during meta-in-context learning. To test this hypothesis, we plotted the proportion of GPT-3's outputs that fell at the extreme values of the target space or outside of it in Figure 4B. We found that meta-in-context learning substantially reduces the number of predictions that are outside of this range, indicating that meta-in-context learning helps to learn a better initial guess.

**Meta-in-context learning works better when task similarity is high:** However, learning a good initial guess is not the full picture. We speculated that meta-in-context learning is more effective if there is a higher similarity to previously encountered tasks. To verify this, we fitted a linear regression model with trial and task similarity on the RMSE. We computed task similarity as the average similarity between the current data-point and the data-points from all previously observed tasks (each individual similarity measure was obtained using a radial basis function kernel). We found a significant effect of trial ($\beta = -0.089 \pm 0.020, t = -9.204, p < 0.001$) as shown in Figure 4C, confirming that performance within a task improves with additional observations. Furthermore, we found a significant effect of task similarity ($\beta = -0.147 \pm 0.020, t = -15.223, p < 0.001$), suggesting that meta-in-context learning works best if similarity to previously encountered tasks is high. In previous experiments, we showed that meta-in-context learning improved performance. This experiment delves into the underlying reasons by proposing how the relatedness of different meta-in-context tasks affects their observed signal. Therefore, we expanded our analysis by also incorporating task similarity considerations to both Experiments 1 and 2 (see Figure 7, Appendix A.2). The analysis reveals the same observation: the more related the task, the more meta-in-context learning is beneficial.

The findings outlined in this section provide further evidence for meta-in-context learning in LLMs.[4] Notably, GPT-3 exhibited superior performance in a real-world multi-dimensional regression task following meta-in-context learning. We identified two reasons for this. First, meta-in-context learning constrained initial model predictions to the range of plausible values, and second, GPT-3 was able to leverage similarities to previously encountered tasks in order to improve its predictions.

## 7 Meta-in-context learning on natural language processing benchmarks

Finally, we examined whether meta-in-context learning also improves upon in-context-learning on standard natural language processing tasks. To test this, we conducted an experiment on the Massive Multitask Language Understanding (MMLU) benchmark [29].

---

[4]Note that while we were running our model simulations, OpenAI released a new model – GPT-4. We repeated the analysis from this section on this new model and described our results in Appendix A.4.

## 7.1 Methods

We focus on the tasks from the STEM supercategory as other supercategories – together with the addition of meta-in-context learning – cause prompt lengths to exceed the limits of GPT-3. For the in-context learning simulations, we provided the model with $k \in \{0, 1, 2\}$ examples from the same category before prompting it on the test question. For the meta-in-context learning simulations, we additionally prepended three examples of two tasks from *different* categories. Examples of two STEM tasks and some extra analysis on open-source models on the entire MMLU can be found in Appendix A.5.

## 7.2 Results

Figure 9 in Appendix A.5 summarizes our results. We found that meta-in-context learning was in general beneficial in terms of performance. The biggest benefit was observed in the zero-shot case, in which meta-in-context learning reached an accuracy of 55.1% outperforming in-context by 22.4%. This illustrates that LLMs do not necessarily have to be prompted by examples from the same category but that they can also transfer some knowledge from different categories.

## 8 Discussion

We have demonstrated that LLMs can improve their in-context learning abilities via in-context learning itself, i.e., that they are capable of *meta-in-context learning*. Meta-in-context learning was not only able to overwrite an LLM's priors but also changed its learning strategies, as demonstrated in two artificial domains. Finally, we applied our approach to two benchmarks. First, a real-world benchmark of regression tasks where we found that meta-in-context learning leads to algorithms that are competitive with standard learning algorithms. Then, we verified the applicability of our results in an NLP benchmark, providing further evidence of the versatility and effectiveness of our approach across diverse contexts.

Perhaps the most significant shortcoming of our model simulations is that they all relied on learning tasks with just a handful of observations. This limitation is mainly due to the practical constraint of a finite context window coupled with meta-in-context learning's rapid prompt length increase. To ensure that we remain within the allowed context length (and to keep the monetary costs for our simulations at a reasonable level), we had to make this design choice. However, we believe that – despite this restriction – our simulations were sufficient to illustrate the potential of meta-in-context learning. We hope that future LLM iterations with longer context lengths and lower inference costs will allow us to extend our simulations to larger datasets.

In addition, the tasks we probed were rather simplistic in their nature. That being said, we think we have reasonably covered the space of fundamental learning paradigms, including a supervised problem, a reinforcement learning problem, a collection of real-world datasets and an NLP benchmark. With the increasing availability of multi-modal models, it will furthermore become feasible to apply our approach to other domains. In this context, two obvious candidates are classification tasks with visual stimuli or grid-based navigation tasks.

A promising direction for further research involves delving into the factors that contribute to the emergence of the meta-in-context learning phenomenon. One potential factor, as mentioned earlier, is the length of the context window. Another avenue to explore is to assess whether specialized models fine-tuned for in-context learning, like the framework proposed by Min et al. [30] (Meta-training for In-Context Learning), yield optimal performance in the context of meta-in-context learning.

In summary, our work has both near- and long-term consequences. In the near term, it indicates that it is not strictly necessary to engineer the perfect prompt for an LLM so that it can solve a given learning problem [31]. Instead, LLMs are – to some degree – able to infer the required information from just a handful of related-tasks examples. In the long term, it could point to a paradigm where we adapt these models to the environment they are applied in purely through meta-in-context learning rather than finetuning them using traditional means.

## Acknowledgments and Disclosure of Funding

We would like to thank Zeb Kurth-Nelson for the helpful discussions. Zeynep Akata acknowledges partial funding by the ERC (853489 - DEXIM).

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
