# A    Appendix[5]

## A.1    Expansion on learning one-dimensional functions

### A.1.1    Meta-in-context learning in open-source models

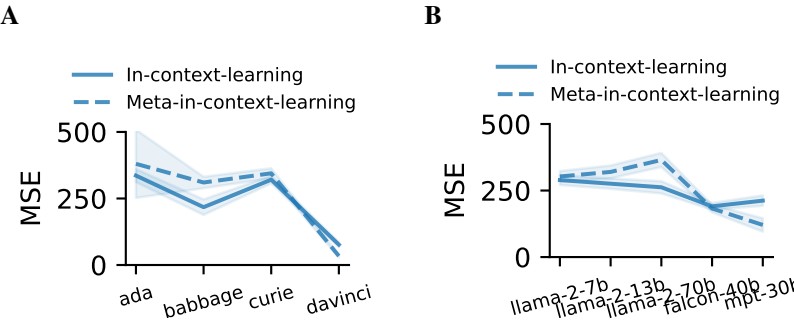

Figure 5: Meta-in-context learning on the linear function learning experiment for multiple LLMs. **A**: Performance of models from the GPT-3 family with increasing model and training sizes. **B**: Performance of various open-source models.

### A.1.2    Meta-in-context learning with non-linear functions

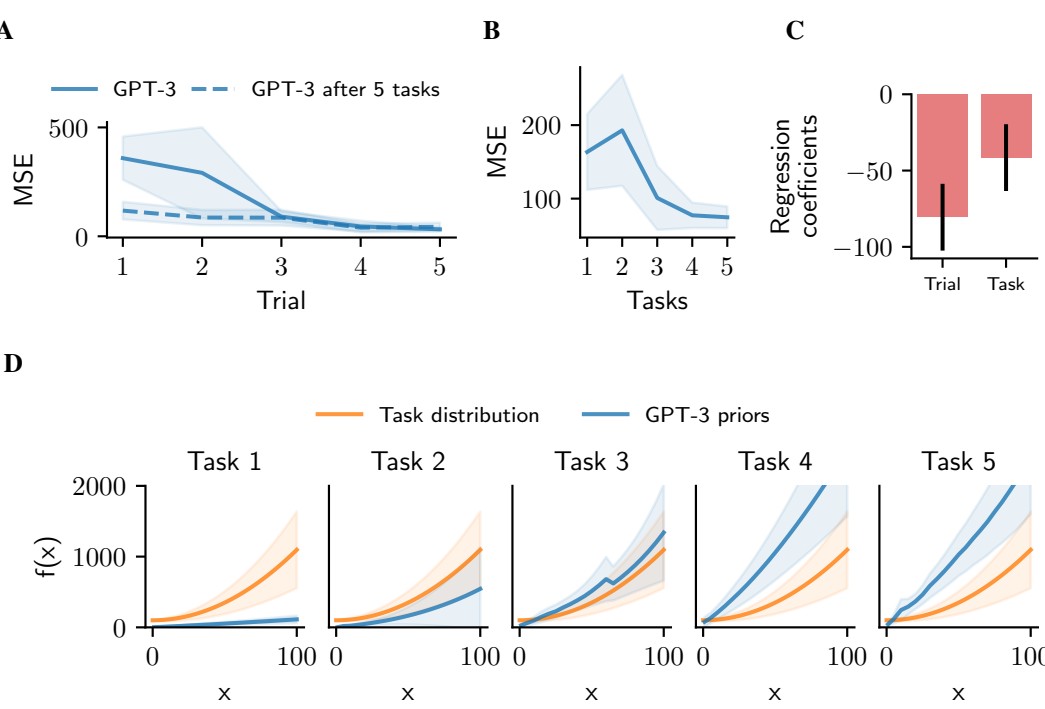

Figure 6: Meta-in-context learning on the one-dimensional regression task with quadratic functions. **A**: MSE across trials. **B**: GPT-3's MSE averaged over trials for each task. **C**: Effects of trial and task for estimating the MSE. **D**: GPT-3's prior expectations across tasks (blue) compared to the true task distribution (orange).

---

[5]Code website: https://github.com/juliancodaforno/meta-in-context-learning

## A.2 Tasks similarity drives meta-in-context learning

We expanded our analysis by incorporating task similarity considerations, extending our approach initially adopted for Experiment 3 (regression on real-world data) to both Experiment 1 (function learning) and Experiment 2 (two-armed bandit). We added task similarity as a regressor on Experiment 1's and 2's respective MSE/regret regression bar plots (Figures 7A and 7B).

For Experiment 1, we quantified task similarity using the average negative L2 norm of the underlying parameters (slope & intercept) with previous tasks. For Experiment 2, we quantified task similarity using the average difference of mean rewards with previous tasks. Our analysis shows a strong effect of task similarity for each Experiment. Interestingly, for the function learning experiment, it seems like all of the task effect gets cancelled when adding a task similarity regressor. This suggests that in this experiment, only similar tasks reduce the MSE.

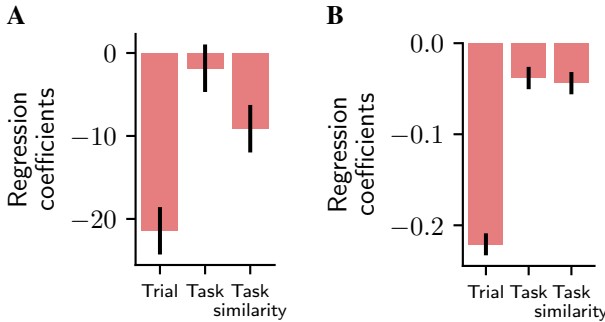

Figure 7: : Effects of trial, task, and task similarity for estimating the MSE or regret. **A**: Function learning experiment. **B**: Two-armed bandit experiment.

### A.3 Details on probit regression analysis for the two-armed bandit experiment

We assume that the agent's beliefs over expected rewards at trial $t$ and action $a$ are captured by the normal distribution $p(r_{a,t}) = \mathcal{N}(\mu_{a,t}, \sigma_{a,t})$[6]. We have relied on the following probit regression analysis proposed by Gershman [26] to investigate which exploration strategies GPT-3 applies when interacting with two-armed bandit problems. This analysis assumes that the agent makes decisions based on three features: value difference $V_t$, relative uncertainty $RU_t$, and value difference divided by total uncertainty $V_t/TU_t$. Formally, these quantities are defined as follows:

$$V_t = \mu_{0,t} - \mu_{1,t}$$
$$RU_t = \sigma_{0,t} - \sigma_{1,t}$$
$$TU_t = \sqrt{\sigma_{0,t}^2 + \sigma_{1,t}^2}$$

where $\mu_{a,t}$ and $\sigma_{a,t}$ represent the agent's beliefs about the mean reward and its corresponding uncertainty estimate for a given arm $a$ at trial $t$. We compute these values using a sequential application of Bayesian inference, assuming normally-distributed priors and likelihoods (updates are only performed for the selected arm):

$$\mu_{a,t+1} = \mu_{a,t} + \alpha_t \left( r_t - \mu_{a,t} \right)$$
$$\sigma_{a,t+1}^2 = \sigma_{a,t}^2 - \alpha_t \sigma_{a,t}^2$$
$$\alpha_t = \frac{\sigma_{a,t}^2}{\sigma_{a,t}^2 + \tau^2}$$

where $\tau$ corresponds to the additive observation noise.

The resulting features are then entered into a probit regression model whose parameters are fit to agent choices via a maximum likelihood estimation using the statsmodels library [32]. We additionally include an interaction effect with task number $k$ for each feature to investigate how exploration behavior changes across tasks. The final model thus contains six features:

$$p(a_t = 0|\mathbf{w}) = \mathbf{\Phi}\left(\mathbf{w}_1 V_t + \mathbf{w}_2 RU_t + \mathbf{w}_3 V_t/TU_t + \mathbf{w}_4 k V_t + \mathbf{w}_5 k RU_t + \mathbf{w}_6 k V_t/TU_t\right)$$

This probit model subsumes several well-known exploration strategies for specific settings of its parameters:

1. Boltzmann-like exploration for $\mathbf{w} = [\mathbf{w}_1, 0, 0, 0, 0, 0]$. Note: Boltzmann exploration would use the logit function instead of the probit. However, the two can used to closely approximate each other: $\sigma(a) \simeq \mathbf{\Phi}(\sqrt{\frac{\pi}{8}}a)$.
2. a noisy version of the UCB algorithm for $\mathbf{w} = [\mathbf{w}_1, \mathbf{w}_2, 0, 0, 0, 0]$.
3. Thompson sampling for $\mathbf{w} = [0, 0, 1, 0, 0, 0]$.

For an exact derivation, see Gershman [26].

---

[6]We obtained these distributions by running Kalman filtering equations on the previously observed data.

## A.4 Regression on real-world data using GPT-4

In this section we investigated the behavior of the newly released GPT-4 model for our last experiment. We proceeded in the same way as for GPT-3.[7] We observed that GPT-4 actually performs slightly worse both before and after meta-in-context learning as shown in Figure 8A. Furthermore, we observed a slightly higher percentage of extreme predictions, particularly for GPT-4's first trial (see Figure 8B). Finally, our analysis also revealed significant effects for trial ($\beta = -0.133 \pm 0.024, t = -10.858, p < 0.001$) and task similarity ($\beta = -0.196 \pm 0.024, t = -15.963, p < 0.001$) as shown in Figure 8C.

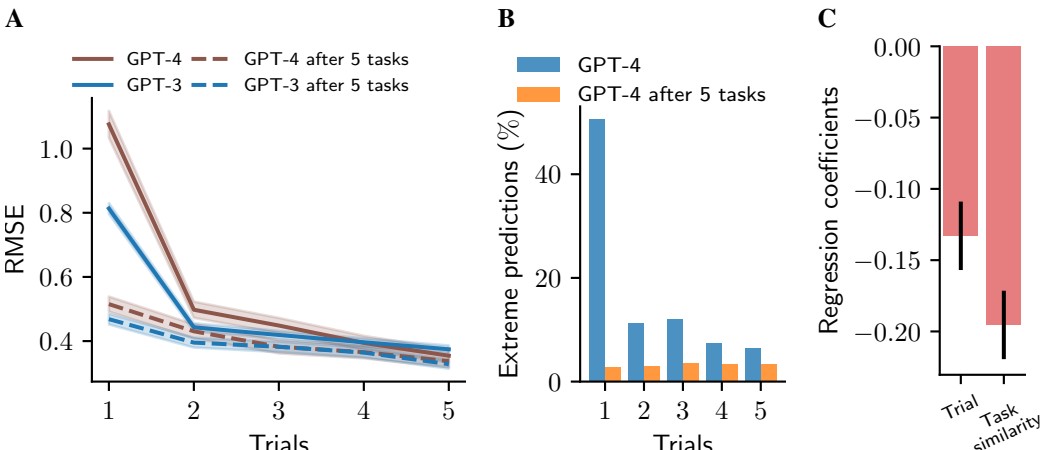

Figure 8: Meta-in-context learning on the regression on real-world data experiment ($42 \cdot 30$ simulations). Errors bars represent $95\%$ confidence intervals. **A**: RMSE across trials for different models. **B**: Percentage of predictions outside or equal to the extremes of the squashed target range. **C**: Effects of trial and task similarities for estimating the RMSE.

---

[7]It is worth noting that the API slightly changed and now provides the option to tailor the message with an *assistant* and a *system*. We did not use them except for the first trial where GPT-4 struggled to give a numerical output. Indeed, for some examples it instead produced messages such as "unfortunately without any information about the relationship between the variables, the prediction is not possible." Therefore, only for that one case, we added the system functionality as follows: {"role": "system", "content": "If no previous examples, sample y from your prior distribution. But do not give any non numerical answer! Even if you are unsure, try to predict y as well as possible."}.

## A.5 MMLU

### A.5.1 Example of two tasks from the STEM category of the MMLU benchmark:

1. The following are multiple choice questions (with answers) about abstract algebra.

    Find all $c$ in $Z_3$ such that $Z_3[x]/(x^2 + c)$ is a field.

    **A.** 0

    **B.** 1

    **C.** 2

    **D.** 3

    Answer: B

    Statement 1 | If $aH$ is an element of a factor group, then $|aH|$ divides $|a|$.
    Statement 2 | If $H$ and $K$ are subgroups of $G$ then $HK$ is a subgroup of $G$.

    **A.** True, True

    **B.** False, False

    **C.** True, False

    **D.** False, True

    Answer: B

    Statement 1 | Every element of a group generates a cyclic subgroup of the group.
    Statement 2 | The symmetric group $S_{10}$ has 10 elements.

    **A.** True, True

    **B.** False, False

    **C.** True, False

    **D.** False, True

    Answer: C

2. The following are multiple choice questions (with answers) about anatomy.

    What is the embryological origin of the hyoid bone?

    **A.** The first pharyngeal arch

    **B.** The first and second pharyngeal arches

    **C.** The second pharyngeal arch

    **D.** The second and third pharyngeal arches

    Answer: D

    Which of these branches of the trigeminal nerve contain somatic motor processes?

    **A.** The supraorbital nerve

    **B.** The infraorbital nerve

    **C.** The mental nerve

    **D.** None of the above

    Answer: D

    A lesion causing compression of the facial nerve at the stylomastoid foramen will cause ipsilateral:

    **A.** paralysis of the facial muscles.

    **B.** paralysis of the facial muscles and loss of taste.

    **C.** paralysis of the facial muscles, loss of taste and lacrimation.

    **D.** paralysis of the facial muscles, loss of taste, lacrimation and decreased salivation.

    Answer:

### A.5.2 MMLU results

We focused on the STEM supercategory of the MMLU benchmark due to limitations imposed by context size (other supercategories contained significantly longer questions) and budget constraints related to the text-davinci-002 engine. Figure 9.A displays the performance of both in-context learning and meta-in-context learning as discussed in Section 7. Additionally, we introduced matched in-context learning where in-context learning involved the same number of examples as meta-in-context learning but all from the same context. For example, six in-context questions on abstract algebra were used, in contrast to six questions spanning various STEM categories preceding a question on abstract algebra. This setup acted as an upper bound for meta-in-context learning, under the extreme scenario where preceding tasks are as similar as those within the same category. The results underscored the effectiveness of meta-in-context learning in NLP tasks.

However, one might argue that this analysis might not allow for sufficient differentiation between tasks. To address this concern, we expanded our evaluation to the entire MMLU benchmark. This enabled a comparable assessment of task similarity, akin to our earlier experiments. Due to the aforementioned constrained on text-davinci-002, we chose to use two open-source models with larger context sizes, namely mpt-30b and Falcon-40b. Task names were assigned to the text-embedding-ada-002, and task similarity was computed by averaging the negative L2 norm between the embedding of the current task name and its preceding task names' embeddings. In both cases, we observed a significant effect ($\beta = 0.1918 \pm 0.076$ and $\beta = 0.1718 \pm 0.077$, both with $p < 0.05$) of task similarity on performance (see Figures 9B and 9C). This suggests that meta-in-context learning does not only leverage the context but also the relationship to the task's context even for NLP tasks as we would expect in a meta-learning framework.

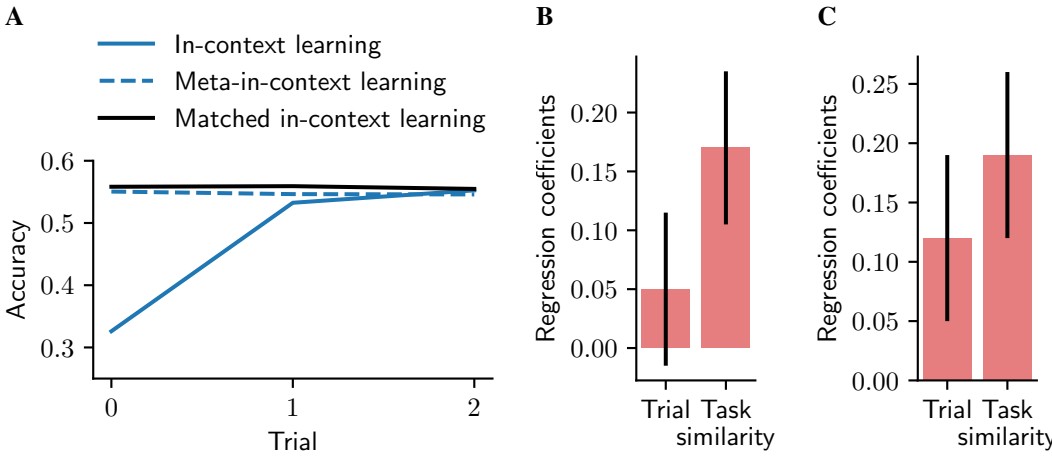

Figure 9: Meta-in-context learning results on the MMLU benchmark. **A**: text-davinci-002 performance on the STEM category. **B**: Effects of trial and task similarities for estimating the accuracy of the mpt-30b LLM on the entire MMLU. **C**: Effects of trial and task similarities for estimating the accuracy of the mpt-30b LLM on the entire MMLU.