# OpenReview forum: "Meta-in-context learning in large language models"
_NeurIPS.cc/2023/Conference — NeurIPS 2023 poster_

### Official Review · Reviewer_6CRk · 2023-07-04

**Soundness:** 2 fair
**Presentation:** 3 good
**Contribution:** 2 fair
**Rating:** 5
**Confidence:** 3

**Summary:**

The paper studies an ability of LLMs, called meta-in-context learning, which showcases that LLMs can recursively improve their in-context learning with demonstrations. The authors illustrate this capability with a regression task and a two-armed bandit task. The analysis demonstrates that LLMs are not only able to learn the task (underlying function) from the examples in this task but also can leverage the examples from other tasks (underlying functions).

**Strengths:**

1. The paper highlights the meta-in-context learning capability of LLMs. The capability allows LLMs to be recursively improved via in-context learning with more tasks in context.
2, The paper is clearly written and easy to follow.

**Weaknesses:**

1. The analysis in the paper demonstrates that LLMs can leverage the examples from other tasks, however, the definition of a task appears to be rather limited in this paper. In the linear regression experiment, for example, a task represents the parameters within a linear function.

2. The authors introduce the concept of meta-in-context learning as a novel ability of LLMs, but it remains unclear how this differs from the traditional in-context learning ability. In this paper, each regression function is regarded as a task with a few examples following the function, and LLMs are found to be able to learn from other tasks. However, this ability seems to have been demonstrated in various other applications. For instance, LLMs can learn to generate dialog responses by observing example responses from other dialogs. If we apply the terminology used in this paper, each dialog can also be considered a task, with each individual utterance serving as an example.

While I think the analysis in the regression and two-armed bandit tasks is commendable, it would be valuable to see such analysis on more realistic tasks, such as dialog generation (a dialog is a task),  and passage/image question answering (a passage or image is a task).

**Questions:**

1. On line 118, it is mentioned that the GPT-3 is not told that underlying functions are linear, distinct from BLR. However, this raises the question of whether GPT-3 possesses the capability to handle non-linear functions. Additionally, if GPT-3 sees a few tasks with linear functions, is GPT-3 able to generalize to a non-linear function in a new task? The meta-in-context learning ability is more valuable if the tasks can be more different.

---

> ### Author Rebuttal · Authors · 2023-08-09
>
> Dear Reviewer 6CRk,
> We thank the reviewer for their helpful comments and we have made a response to each of their comments along with suggested changes to the paper:
>
> > 1. The paper highlights the meta-in-context learning capability of LLMs. The capability allows LLMs to be recursively improved via in-context learning with more tasks in context. 2, The paper is clearly written and easy to follow.
> [...] I think the analysis in the regression and two-armed bandit tasks is commendable
>
> We appreciate the reviewer’s comments on the clarity and presentation of the paper as well as exposing the two-armed bandit task as commendable.
>
> > 2. The authors introduce the concept of meta-in-context learning as a novel ability of LLMs, but it remains unclear how this differs from the traditional in-context learning ability. In this paper, each regression function is regarded as a task with a few examples following the function, and LLMs are found to be able to learn from other tasks. However, this ability seems to have been demonstrated in various other applications. For instance, LLMs can learn to generate dialog responses by observing example responses from other dialogs. If we apply the terminology used in this paper, each dialog can also be considered a task, with each individual utterance serving as an example.
>
> Thank you for this comment. We view the relationship between in-context learning and meta-in-context learning as similar to the relationship between Bayesian inference and hierarchical Bayesian inference. They are both based on the same algorithmic principles applied at different conceptual levels. In-context learning and Bayesian inference are used for within-task learning, while meta-in-context learning and hierarchical Bayesian inference are used to pool information across tasks.  Formulating explicit tasks and episodes in the way that we do allows us to quantify the degree of meta-in-context learning, whereas simply having dialogue responses generated following from other dialogue is hard to quantify.
>
> > While I think the analysis in the regression and two-armed bandit tasks is commendable, it would be valuable to see such analysis on more realistic tasks, such as dialog generation (a dialog is a task), and passage/image question answering (a passage or image is a task).
>
> We appreciate this suggestion which was echoed by all other reviewers as well. We have therefore conducted additional simulations of meta-in-context learning on the Massive Multitask Language Understanding (MMLU) benchmark. We will add the following section to our revised paper:
>
> "**Meta-in-context learning on natural language processing benchmarks**:
>
> Finally, we examined whether meta-in-context learning also improves upon in-context-learning on standard natural language processing tasks. To test this, we conducted an experiment on the Massive Multitask Language Understanding (MMLU) benchmark \cite{hendrycks2020measuring}.
>
> **Methods**:
> We focus on the tasks from the STEM supercategory as other supercategories -- together with the addition of meta-in-context learning -- cause prompt lengths to exceed the limits of GPT-3. For the in-context learning simulations, we provided the model with $k \in {0, 1, 2}$ examples from the same category before prompting it on the test question. For the meta-in-context learning simulations, we additionally prepended three examples of two tasks from \emph{different} categories.
>
> **Results**:
> Figure 9 summarizes our results. We found that meta-in-context learning was in general beneficial in terms of performance. The biggest benefit was observed in the zero-shot case, in which meta-in-context learning reached an accuracy of $55.1$ percent outperforming in-context by $22.4$ percent. This illustrates that LLMs do not necessarily have to be prompted by examples from the same category but that they can also transfer some knowledge from different categories."
>
> The figure with the corresponding results can be found in the attached PDF under Figure 9.
>
> > On line 118, it is mentioned that the GPT-3 is not told that underlying functions are linear, distinct from BLR. However, this raises the question of whether GPT-3 possesses the capability to handle non-linear functions. Additionally, if GPT-3 sees a few tasks with linear functions, is GPT-3 able to generalize to a non-linear function in a new task? The meta-in-context learning ability is more valuable if the tasks can be more different.
>
> We appreciate the feedback and agree that additional evaluation on non-linear functions does make the analysis more comprehensive. We have therefore added the following section to our revised paper:
>
> "**Meta-in-context learning with non-linear functions**:
> We also aimed to determine whether GPT-3 could use meta-in-context learning to adapt to non-linear functions. In the Supplementary material, we present an analysis using quadratic functions conducted within the same experimental framework. In summary, we find that GPT-3 indeed exhibits the ability to meta-in-context learn non-linear functions."
>
> For a comprehensive visualization of these refined insights, we refer readers to Figure 7 in the attached PDF.

---

> > ### Comment · Reviewer_6CRk · 2023-08-18
> >
> > Thank you for the response and the efforts to address my concerns. I think the new experiments can better explain the effectiveness of the meta ICL. I still have a question about the results in MMLU. The meta ICL only performs better than ICL for zero-shot, where ICL sees no demonstration examples and meta ICL sees a few examples although they are from another category. Given the small difference between these categories (same task format but different subjects, e.g. physics vs math), it won't be surprising that having examples is better than the zero-shot setting. I have updated my score based on the other experiments but I am looking forward to more elaboration on the MMLU results.

---

> > > ### Author Response · Authors · 2023-08-19
> > >
> > > We greatly appreciate the feedback provided by the reviewer, which we believe resonates consistently with their earlier observations about the definition of a task and its distinction from conventional in-context learning. While we acknowledge the similarity between our MMLU results and previous empirical findings for real-world data, as demonstrated by the example dialogue the reviewer provided, we maintain our position that our framework offers a unique quantitative assessment of the extent of meta-in-context learning. This contrasts with previous research, which primarily reported performance improvements without delving into the underlying mechanisms. Below, we hope to convince the reviewer that the performance improvement is not solely coming from a very close task that can be seen as an example but also from how related the task is, even for NLP experiments.
> > >
> > > **Elaboration on MMLU results: Task similarity has a significant effect on accuracy on the MMLU benchmark:**
> > > To elaborate on the benefits of our analysis, we wanted to quantify again the use of task similarity for meta-in-context learning in the MMLU benchmark. As explained before, due to context size and price constraints, the text-davinci-002 engine was only tested on the STEM subcategory of the MMLU benchmark which does not allow enough differentiation between tasks as the reviewer mentioned. Therefore, we ran two open-source models with larger context sizes on the entire MMLU benchmark, namely mpt-30b and Falcon-40b. The task name was given to the text-embedding-ada-002 and the task similarity was computed as the average negative L2 norm between the task name's embedding and its previous task name's embeddings. For both, we see a significant effect ( Beta=0.1918 +- 0.076 & Beta=0.1718 +- 0.077 with both p>0.05). This suggests that meta-in-context learning does not only leverage the context but also the relationship to the task's context even for NLP tasks as we would expect in a meta-learning framework. This analysis will be included in the Supplementary material. If there are any remaining inquiries, we are available for further discussion.

---

> > > > ### Comment · Reviewer_6CRk · 2023-08-21
> > > >
> > > > Thank the authors for answering my question. I have updated my score.

---

### Official Review · Reviewer_NP3F · 2023-07-06

**Soundness:** 3 good
**Presentation:** 2 fair
**Contribution:** 2 fair
**Rating:** 4
**Confidence:** 3

**Summary:**

The authors demonstrate that the in-context learning abilities of large language models can be recursively improved via in-context learning itself.

The paper misses the method section. I don't know the details and cannot tell the difference from previous work.

I didn't get the novel part of the method. It seems an empirical study.

The authors had better do some experiments on benchmark datasets, such as the datasets from GPT-3 paper.

**Strengths:**

1. The problem explored is critical.
2. The experimental analyses are interesing.

**Weaknesses:**

1. Missing model section.
2. The method is more of an empirical study.
3. The experiments are not solid. The authors focus on the GPT-3 model exploration but miss comparison to any benchmark dataset from GPT-3 paper. The shot number, context length, and language understanding in benchmark datasets are all critical issues to study. And GPT-3 is released quite a long time ago. Exploring recent LLMs, such as ChatGPT or GPT-4, would be better.
4. The finding is similar to GPT-3 on domain adaptation, such as the work from "Prompting GPT-3 To Be Reliable".

**Questions:**

Can you formulate meta-in-context learning?

**Limitations:**

I didn't get the novel contribution of the work.

---

> ### Author Rebuttal · Authors · 2023-08-09
>
> Dear Reviewer NP3F,
>
> We thank the reviewer for their helpful comments and we have made a response for each of their comments along with suggested changes to the paper:
>
> > Strengths:
> The problem explored is critical.
> The experimental analyses are interesing.
>
> We thank the reviewer for exposing the research problem as critical and the analyses as interesting.
>
> > Weaknesses:
> Missing model section.
> The method is more of an empirical study.
>
> We thank the reviewer for the feedback. However, we do not view this as a weakness. NeurIPS publishes hundreds of empirical studies every year that do not propose new models.
>
> > The experiments are not solid. The authors focus on the GPT-3 model exploration but miss comparison to any benchmark dataset from GPT-3 paper. The shot number, context length, and language understanding in benchmark datasets are all critical issues to study.”
>
> We appreciate the criticism. In our revised paper, we have conducted additional simulations of meta-in-context learning on the Massive Multitask Language Understanding (MMLU) benchmark. We will add the following section to our revised paper:
>
> "**Meta-in-context learning on natural language processing benchmarks**:
>
> Finally, we examined whether meta-in-context learning also improves upon in-context-learning on standard natural language processing tasks. To test this, we conducted an experiment on the Massive Multitask Language Understanding (MMLU) benchmark \cite{hendrycks2020measuring}.
>
> **Methods**:
> We focus on the tasks from the STEM supercategory as other supercategories -- together with the addition of meta-in-context learning -- cause prompt lengths to exceed the limits of GPT-3. For the in-context learning simulations, we provided the model with $k \in {0, 1, 2}$ examples from the same category before prompting it on the test question. For the meta-in-context learning simulations, we additionally prepended three examples of two tasks from \emph{different} categories.
>
> **Results**:
> Figure 9 summarizes our results. We found that meta-in-context learning was in general beneficial in terms of performance. The biggest benefit was observed in the zero-shot case, in which meta-in-context learning reached an accuracy of $55.1$ percent outperforming in-context by $22.4$ percent. This illustrates that LLMs do not necessarily have to be prompted by examples from the same category but that they can also transfer some knowledge from different categories."
>
> The figure with the corresponding results can be found in the attached PDF under Figure 9.
>
> > And GPT-3 is released quite a long time ago. Exploring recent LLMs, such as ChatGPT or GPT-4, would be better.
>
> We appreciate the feedback and agree that more analysis on more recent LLMs (especially open-source models which are transparent on training and architectures) does improve the empirical analysis. Therefore, we conducted additional simulations of meta-in-context learning on the latest open-source models (MPT-30B, Falcon-40B, Llama-2-7B/13B/70B). We will add the following section to our revised paper:
>
> "**Meta-in-context learning in open-source models**:
> To comprehensively explore meta-in-context learning, we extended our evaluation to encompass five distinct open-source models. This expansion allowed us to examine whether this phenomenon is also applicable to less opaque model architectures. Notably, while the majority of models  (Falcon-40B \cite{falcon40b} and Llama-2 \cite{touvron2023llama2} models) exhibited no indications of meta-in-context learning, intriguingly, MosaicML's MPT-30B \cite{MosaicML2023Introducing} demonstrated this ability. The results are included in the Supplementary material."
>
> The figure with the corresponding results can be found in the attached PDF under Figure 6B.
> We also want to point out to the reviewer that the GPT-4 model also was already included for the real-world regression in the Supplementary materials (see Figure 5).
>
> > The finding is similar to GPT-3 on domain adaptation, such as the work from "Prompting GPT-3 To Be Reliable".
>
> We thank the reviewer for mentioning this paper. We have included a reference to it in our discussion. That being said, we do not think this paper is particularly relevant to our work (besides the fact that both are concerned with in-context learning). While Si et al. propose ways to make in-context learning more reliable using prompt engineering, we show that GPT-3 is capable of adapting its in-context learning abilities to previously encountered related tasks without the need for prompt engineering. In addition, "Prompting GPT-3 to be reliable" focuses on different metrics (4 aspects of reliability), whereas our paper focuses on a different ability (meta-learning).
>
> > Questions:
> Can you formulate meta-in-context learning?
>
> We believe that Figure 1 provides a nice formulation of what we view as meta-in-context learning:
>
> “High-level overview of our approach on an example of multiple three-shot regression tasks. We present an LLM with $N$ learning tasks in a row. Improvement within a task indicates that the model is capable of in-context learning. If in-context learning improves across multiple learning tasks, the model is also capable of meta-in-context learning.”

---

### Official Review · Reviewer_3waD · 2023-07-11

**Soundness:** 4 excellent
**Presentation:** 4 excellent
**Contribution:** 2 fair
**Rating:** 7
**Confidence:** 3

**Summary:**

The authors undertake a study of "meta in-context learning" as a capability of Large Language Models, specifically focused on GPT-3 (with some initial experiments on GPT-4). The authors define meta in-context learning following a task-trial structure, in which the agent observes multiple tasks each consisting of multiple trials. In each trial, a given input-output pair is observed; on the final trial, the model needs to predict the output for an unpaired input. The (unobserved) function generating outputs from inputs differs between tasks. The authors show that GPT-3 is able to learn not only at the trial level (traditional in-context learning) but also across tasks; that is, it can get better both at modelling a specific function and at the general task of modelling functions (in a particular context). The authors further show that this holds across artificial simple regression, reinforcement learning and multiple regression.on real-world data.

**Strengths:**

The paper is well-written, and the results are thoroughly investigated and clearly conveyed. The authors seek to identify the specific causes of the behaviours they document and the extent to which they can be attributed to simpler processes, e.g. learning the output distribution for predicting on the first task of a trial. The paper includes a discussion of context window size-related limitations to the applicability of the results.

On originality, I don't know of any other works that address the question of meta-learning without weight updates. As the authors point out, the results could inspire other researchers and applied LLM users to persue multi-task setups rather than finetuning.

**Weaknesses:**

The paper is well-reasoned within the specific niche of tasks demonstrated, though the overall implications of the work are arguable. Because the examples are constrained to simple numerical problems, it's difficult to tell whether the capability can be usefully leveraged on other types of language tasks. In the discussion, where it's argued that meta in-context could replace finetuning, it would help to give a few examples of real-world tasks that could be reframed as meta in-context.

Within the artificial tasks, it would also be nice to see non-numerical demonstrations of meta in-context learning, both to show that it is possible on other types of tasks and because LLMs tend to struggle with numerical transformations. For instance, within-task could involve translating sentences in a simple made-up language into English (with words that are repeated across trials), while each task involves a different made-up language. This could provide a more convincing argument that meta in-context learning is a general phenomenon beyond math tasks.

More detail on the exploration probit regression model and investigated regression coefficients in the Supplementary would be useful.

**Questions:**

Is it possible to create a clear definition of what constitutes an in-context trial and what constitututes an across-context trial, i.e. where does in-context learning become meta in-context learning? Does an intial task trial always start with some degree of irreduceable uncertainty?

See the above questions about real-world examples or non-numerical examples of meta in-context learning.

**Limitations:**

No major negative societal implications are expected. The authors address a number of limitations, including low sample size and simplicity of the tasks, in the Discussion.

---

> ### Author Rebuttal · Authors · 2023-08-09
>
> Dear Reviewer 3waD,
>
> We thank the reviewer for finding the results thoroughly investigated and clearly conveyed. We also appreciate the comment on originality and belief in its potential impact for applying LLMs to multi-task setups rather than fine-tuning. We also thank the reviewer for their helpful comments and we have made a response for each of their comments along with suggested changes to the paper.
>
> > The paper is well-reasoned within the specific niche of tasks demonstrated, though the overall implications of the work are arguable. Because the examples are constrained to simple numerical problems, it's difficult to tell whether the capability can be usefully leveraged on other types of language tasks. [...]
> Within the artificial tasks, it would also be nice to see non-numerical demonstrations of meta in-context learning, both to show that it is possible on other types of tasks and because LLMs tend to struggle with numerical transformations.
>
> We appreciate this suggestion which was echoed by all other reviewers as well. We have therefore conducted additional simulations of meta-in-context learning on the Massive Multitask Language Understanding (MMLU) benchmark. We will add the following section to our revised paper:
>
> "**Meta-in-context learning on natural language processing benchmarks**:
>
> Finally, we examined whether meta-in-context learning also improves upon in-context-learning on standard natural language processing tasks. To test this, we conducted an experiment on the Massive Multitask Language Understanding (MMLU) benchmark \cite{hendrycks2020measuring}.
>
> **Methods**:
> We focus on the tasks from the STEM supercategory as other supercategories -- together with the addition of meta-in-context learning -- cause prompt lengths to exceed the limits of GPT-3. For the in-context learning simulations, we provided the model with $k \in {0, 1, 2}$ examples from the same category before prompting it on the test question. For the meta-in-context learning simulations, we additionally prepended three examples of two tasks from different categories.
>
> **Results**:
> Figure 9 summarizes our results. We found that meta-in-context learning was in general beneficial in terms of performance. The biggest benefit was observed in the zero-shot case, in which meta-in-context learning reached an accuracy of $55.1$ percent outperforming in-context by $22.4$ percent. This illustrates that LLMs do not necessarily have to be prompted by examples from the same category but that they can also transfer some knowledge from different categories."
>
> The figure with the corresponding results can be found in the attached PDF under Figure 9.
>
> > More detail on the exploration probit regression model and investigated regression coefficients in the Supplementary would be useful.
>
> We agree with the reviewer that more details on this model would be helpful to the reader. We have therefore added a section to the Supplementary Materials called “Details on probit regression analysis” which describes the corresponding analysis in more detail:
>
> "**Details on probit regression analysis**:
>
> We have relied on a probit regression analysis proposed by Gershman \cite{gershman2018deconstructing} to investigate which exploration strategies GPT-3 applies when interacting with two-armed bandit problems. This analysis assumes that the agent makes decisions based on three features: value difference $V_t$, relative uncertainty $RU_t$, and value difference divided by total uncertainty $V_t/TU_t$. Formally, these quantities are defined as follows:
>
> EQUATION
>
> where $\mu_{a, t}$ and $\sigma_{a, t}$ represent the agent’s beliefs about the mean reward and its corresponding uncertainty estimate for a given arm $a$ at trial $t$. We compute these values using via a sequential application of Bayesian inference, assuming normally-distributed priors and likelihoods (updates are only performed for the selected arm):
>
> EQUATION
>
> where $\tau$ corresponds to the additive observation noise.
> The resulting features are then entered into a probit regression model whose parameters are fit to agent choices via a maximum likelihood estimation using the statsmodels library \cite{seabold2010statsmodels}. We additionally include an interaction effect with task number $k$ for each feature to investigate how exploration behavior changes across tasks. The final model thus contains six features:
>
> EQUATION
>
> This probit model subsumes several well-known exploration strategies for specific settings of its parameters:
>
> 1. Boltzmann-like exploration for $\mathbf{w} = [\mathbf{w}_{1}, 0, 0, 0, 0, 0]$. Note: Boltzmann exploration would use the logit function instead of the probit. However, the two can used to closely approximate each other: $\sigma(a) \simeq \mathbf{\Phi}(\sqrt{\frac{\pi}{8}}a)$.
> 2. a noisy version of the UCB algorithm for $\mathbf{w} = [\mathbf{w}_1, \mathbf{w}_2, 0, 0, 0, 0]$.
> 3. Thompson sampling for $\mathbf{w} = [0, 0, 1, 0, 0, 0]$. For an exact derivation, see Gershman \cite{gershman2018deconstructing}."
>
> > Is it possible to create a clear definition of what constitutes an in-context trial and what constitututes an across-context trial, i.e. where does in-context learning become meta in-context learning? Does an intial task trial always start with some degree of irreduceable uncertainty?
>
> Thank you for this comment. We view the relationship between in-context learning and meta-in-context learning as similar to the relationship between Bayesian inference and hierarchical Bayesian inference. They are both based on the same algorithmic principles applied at different conceptual levels. In-context learning and Bayesian inference are used for within-task learning, while meta-in-context learning and hierarchical Bayesian inference are used to pool information across tasks.

---

> > ### Comment · Reviewer_3waD · 2023-08-13
> > **Re: Rebuttal**
> >
> > Thank you to the authors for the response, updates and undertaking the new analyses, particularly the MMLU analysis, which make me lean more confidently towards acceptance. I will review and consider raising my score.

---

> > > ### Author Response · Authors · 2023-08-13
> > >
> > > We appreciate the reviewer for dedicating their time to our response and for their kind words. If there are any remaining inquiries, we are available for further discussion.

---

> > > > ### Comment · Reviewer_3waD · 2023-08-21
> > > > **Re: MMLU**
> > > >
> > > > I am updating my score to a 7 with the addition of the MMLU. While the theoretical contribution is still somewhat limited, I think the paper will provide useful insight to other researchers and practitioners looking to leverage in-context learning in their own work.
> > > >
> > > > One last question: could you provide example an example in-context and meta-in-context prompt from the MMLU task (both for the paper and for us as reviewers, if possible)? I'm particularly interested in the degree of overlap between different categories in the STEM supercategory.

---

> > > > > ### Author Response · Authors · 2023-08-21
> > > > >
> > > > > We appreciate the feedback and agree that additional clarity on the MMLU benchmark for the reader will be helpful. In fact, reviewer 6CRkc was also curious about task overlap. If it is of interest, we ran two open-source models with longer context sizes to allow for runs on the entire MMLU benchmark; we show a statistically significant effect of task similarity on meta-in-context learning performance (you can find more details in our discussion with reviewer 6CRk).
> > > > >
> > > > > In terms of an example of two STEM tasks, please find them (abstract algebra & anatomy) below. We will add it to the Supplementary Material. It is worth noting that the only difference between meta-in-context learning and the standard in-context learning evaluation is the concatenation of tasks. If there are any remaining inquiries, we are available for further discussion.
> > > > >
> > > > > **Example of two tasks:**
> > > > >
> > > > > **1:**
> > > > >
> > > > > The following are multiple choice questions (with answers) about abstract algebra.
> > > > > Find all c in Z_3 such that Z_3[x]/(x^2 + c) is a field.
> > > > > A. 0
> > > > > B. 1
> > > > > C. 2
> > > > > D. 3
> > > > > Answer: B
> > > > >
> > > > > Statement 1 | If aH is an element of a factor group, then |aH| divides |a|. Statement 2 | If H and K are subgroups of G then HK is a subgroup of G.
> > > > > A. True, True
> > > > > B. False, False
> > > > > C. True, False
> > > > > D. False, True
> > > > > Answer: B
> > > > >
> > > > > Statement 1 | Every element of a group generates a cyclic subgroup of the group. Statement 2 | The symmetric group S_10 has 10 elements.
> > > > > A. True, True
> > > > > B. False, False
> > > > > C. True, False
> > > > > D. False, True
> > > > > Answer: C
> > > > >
> > > > > **2:**
> > > > >
> > > > > The following are multiple choice questions (with answers) about anatomy.
> > > > > What is the embryological origin of the hyoid bone?
> > > > > A. The first pharyngeal arch
> > > > > B. The first and second pharyngeal arches
> > > > > C. The second pharyngeal arch
> > > > > D. The second and third pharyngeal arches
> > > > > Answer: D
> > > > >
> > > > > Which of these branches of the trigeminal nerve contain somatic motor processes?
> > > > > A. The supraorbital nerve
> > > > > B. The infraorbital nerve
> > > > > C. The mental nerve
> > > > > D. None of the above
> > > > > Answer: D
> > > > >
> > > > > A lesion causing compression of the facial nerve at the stylomastoid foramen will cause ipsilateral
> > > > > A. paralysis of the facial muscles.
> > > > > B. paralysis of the facial muscles and loss of taste.
> > > > > C. paralysis of the facial muscles, loss of taste and lacrimation.
> > > > > D. paralysis of the facial muscles, loss of taste, lacrimation and decreased salivation.
> > > > > Answer:

---

### Official Review · Reviewer_xRJY · 2023-07-20

**Soundness:** 3 good
**Presentation:** 2 fair
**Contribution:** 2 fair
**Rating:** 5
**Confidence:** 4

**Summary:**

The paper explores a phenomenon referred to as “meta in-context learning” in large language models, where the in-context learning abilities of these language models can be recursively enhanced through in-context learning itself. To demonstrate this, the researchers examine two idealized domains: a one-dimensional regression task and a two-armed bandit task. They show that meta-in-context learning (1) improves in-context learning performance (2) dynamically shapes the large language model's priors over expected tasks and (3) also modifies its in-context learning strategies.


**Strengths:**

* The high level question raised in this paper is interesting – can in-context learning be improved via the recursive operation of concatenating several different tasks in-context?
* The experimental setup is clean and the obtained experimental results convey a clear message. They clearly show that for the examined problems meta in-context learning helps, and moreover they reveal that the improvement (also) comes from previous tasks in context shaping the LLM prior for the examined task.




**Weaknesses:**

* I found the setup and particularly the examined tasks to be too limited. Although the high level question is interesting I feel like much more could be done in order to shed light on it and connect it to actual scenarios in which in-context learning is used in practice.
The authors chose to focus on linear regression (synthetic and 60 real world tasks filtered down to 42) / synthetic bandit problems. The abstract and intro describe that in-context learning is a very important feature of LLMs which facilitated their success; while true it is not related to these types of tasks but rather to natural language ones. I think that the interesting high level question raised in this paper could easily have been studied for natural language tasks, and in particular relating it to papers showing multitask benefits which study which tasks are related and reinforce each other such as Sanh et al ICLR 2022, Aribandi et al, ICLR 2022 and others.
I acknowledge that there is a growing literature of works theoretically analyzing and experimenting with in-context learning on synthetic / mathematical problems, the authors mention some of it. However, in this case I think that the merits of the synthetic / simple / mathematic setup were not fully utilized – for example, the authors could have studied in this simple setting how the relatedness of different tasks in the same context affects their observed signal (eg by defining a formal metric of distance / similarity between linear regression tasks). A small subset of the results in section 3.3 is dedicated to a statement about task similarity and I think it should be expanded for deeper insight. In the lack of such deeper investigations in the synthetic realm, and on the other hand no experimentation on language tasks, this paper misses out on real insight relevant for language related scenarios. For the scenarios studied in the paper, LLMs are not the go-to tool. I will note as an outlier to the above that the study of  in-context learning strategies in the bandit case is deep, novel, and leverages the clean mathematical structure of the synthetic experiment. However, even in this case, I see no conclusions applicable to real world scenarios in which LLM in-context learning is the go-to tool.

* The paper is not segmented into “method”, “experimental setup”, “results” sections, but rather includes one section describing all of the above linearly according to the different tasks that were ran. This is not a problem a-priori, but I found the presentation hard to follow at times, and the authors may want to consider structuring the paper a bit more.

* Minor remarks:
** There are some some typos (eg, end of parentheses in line 77, named citations without a year)
** The name “meta in-context learning” is used for a different method presented by Min et al. 2021 in this influential paper: “MetaICL: Learning to Learn In Context”. You may want to rename this paper.
** I think that the authors overstate when they claim meta in-context learning to be related to psychological experiments.




**Questions:**

What are the main practical consequences or implications of your findings for specific real world scenarios in which in-context learning in LLMs is currently used?

**Limitations:**

Yes

---

> ### Author Rebuttal · Authors · 2023-08-09
>
> Dear Reviewer xRJY,
>
> First, we thank the reviewer for finding the paper interesting with a clear message conveyed from the experiments. We also thank the reviewer for their helpful comments and we have made a response for each of their comments along with suggested changes to the paper:
>
> > I found the setup and particularly the examined tasks to be too limited. [...] I think that the interesting high level question raised in this paper could easily have been studied for natural language tasks.
>
> We appreciate this suggestion which was echoed by all other reviewers as well. We have therefore conducted additional simulations of meta-in-context learning on the Massive Multitask Language Understanding (MMLU) benchmark. We will add the following section to our revised paper:
>
> "**Meta-in-context learning on natural language processing benchmarks**:
> Finally, we examined whether meta-in-context learning also improves upon in-context-learning on standard natural language processing tasks. To test this, we conducted an experiment on the Massive Multitask Language Understanding (MMLU) benchmark \cite{hendrycks2020measuring}.
>
> **Methods**:
> We focus on the tasks from the STEM supercategory as other supercategories -- together with the addition of meta-in-context learning -- cause prompt lengths to exceed the limits of GPT-3. For the in-context learning simulations, we provided the model with $k \in {0, 1, 2}$ examples from the same category before prompting it on the test question. For the meta-in-context learning simulations, we additionally prepended three examples of two tasks from \emph{different} categories.
>
> **Results**:
> Figure 9 summarizes our results. We found that meta-in-context learning was in general beneficial in terms of performance. The biggest benefit was observed in the zero-shot case, in which meta-in-context learning reached an accuracy of $55.1$ percent outperforming in-context by $22.4$ percent. This illustrates that LLMs do not necessarily have to be prompted by examples from the same category but that they can also transfer some knowledge from different categories."
>
> The figure with the corresponding results can be found in the attached PDF under Figure 9.
>
> >  The authors could have studied in this simple setting how the relatedness of different tasks in the same context affects their observed signal (eg by defining a formal metric of distance /similarity between linear regression tasks).
>
> We agree that beyond merely showcasing performance gains across tasks, our paper would benefit from delving into the underlying reasons for the improved performance of LLMs across tasks. In response, we have expanded our analysis by incorporating task similarity considerations, extending our approach initially adopted for Experiment 3 to both Experiment 1 and Experiment 2. We added task similarity as a regressor on Experiment 1’s and 2’s respective MSE/regret regression bar plots (Figure 2C and 3C).
>
> For Experiment 1, we quantified task similarity using the average negative L2 norm of the underlying parameters (slope & intercept) with previous tasks. For Experiment 2, we quantified task similarity using the average difference of mean rewards with previous tasks. Our analysis shows a strong effect of Task similarity for each Experiment.
>
> For a comprehensive visualization of these refined insights, we refer readers to the attached PDF which also includes the updated barplots in Figure 8.
>
> > I see no conclusions applicable to real world scenarios in which LLM in-context learning is the go-to tool.
>
> We thank the reviewer for raising the point regarding conclusions applicable to real-world scenarios. We agree that adding more real-world scenarios would further strengthen our paper. To address this point, we have added experiments on a natural language benchmark (MMLU) as described in more detail in our earlier response above.
>
> > The paper is not segmented into “method”, “experimental setup”, “results” sections.
>
> We thank the reviewer for this feedback, which we have used to restructure the sections of the paper. In particular, we have moved each experimental subsection one level higher (e.g., “3.1 Learning one-dimensional functions” is now “4 Learning one-dimensional functions”). Furthermore, each of these sections is now divided into a methods and results subsection. Finally, we have renamed “3 Experimental analyses” to “3 Experimental setup”. We hope that these changes make it easier for readers to follow our presentation.
>
> > Minor remarks:
> ** There are some some typos
> ** The name “meta in-context learning” is used for a different method presented by Min et al. 2021 in this influential paper: “MetaICL: Learning to Learn In Context”. You may want to rename this paper.
> ** I think that the authors overstate when they claim meta in-context learning to be related to psychological experiments.
>
> We addressed the typos and agree that the link to psychological experiments is not clear so we removed the paragraph from our paper.
>
> In addition, we want to thank the reviewer for mentioning the MetaICL paper. We have added it to the related work section, where we also describe how it differs from our approach:
>
> "Meta-in-context learning versus classical meta-learning schemes:
> [...] **Min et al. \cite{min2021metaicl} proposed a method called \emph{meta-training for in-context learning} following the same paradigm. Their work starts with a pretrained language model that is then trained (via an adjustment of its weights) on a large set of training tasks to do in-context learning, ultimately leading to a model with improved in-context learning capabilities.** In contrast to these approaches, our approach adapts a model to a series of learning problems entirely through the context itself as opposed to updating weights. "
>
> Note that MetaICL stands for “meta-training for in-context learning” and not for “meta-in-context learning”, so we do not think that an entire renaming of our terminology is necessary.

---

> > ### Comment · Reviewer_xRJY · 2023-08-14
> >
> > I thank the authors for their thorough rebuttal. I looked also at the graphs in the supplementary PDF. The zero-shot meta-ICL results on MMLU are encouraging. I am raising my score and I highly recommend expanding on these experiments if the paper is published at NeurIPS or when submitting the next venue.

---

> > > ### Author Response · Authors · 2023-08-15
> > >
> > > We extend our gratitude to the reviewer for investing their time in reviewing our response and for their generous remarks. We acknowledge the feedback and, as a response, we are broadening our experimentation on the MMLU benchmark. This expansion involves incorporating various open-source models, and the outcomes of these experiments will be included in the Supplementary material. If there are any remaining inquiries, we are available for further discussion.

---

### Official Review · Reviewer_ShPz · 2023-07-26

**Soundness:** 3 good
**Presentation:** 4 excellent
**Contribution:** 3 good
**Rating:** 5
**Confidence:** 4

**Summary:**

This paper demonstrates that large language models (LLMs) are capable of meta-in-context learning: updating their in-context-learning abilities when prompted with examples of several tasks
The authors empirically show this capability on several learning paradigms, including supervised learning (1D linear regression), reinforcement learning (2-armed bandits), and several real-world linear regression problems.
These experiments show that LLMs, and GPT-3/GPT-4 more particularly, can effectively adapt their in-context learning algorithm while also matching simple baselines like Bayesian linear regression (BLR), upper-confidence bound (UCB), and random forests.
They also show that meta-in-context can outperform standard in-context learning, which is especially relevant given the popularity of this technique with LLMs.

**Strengths:**

1. I found the paper an enjoyable read. The research is relatively well motivated, the analysis are carefully detailed, and the method is clearly explained. In fact, I believe I could easily replicate the results from the paper because all experimental testbeds are precisely described and the exact prompts are provided (which also help illustrate the method; see blue panels). While more baselines could be included for completeness, the experiments clearly show the meta-in-context learning effect so I don't think they are necessary.
2. Despite its simplicity, the presented idea is elegant and I could see it impacting research beyond large language models. For exampe it is not difficult to imagine extensions beyond text inputs (say, vision, audio, or robotics), or theory extensions (in the flavor of "Transformers learn in-context by gradient descent"), or even applying a similar strategy beyond in-context learning (eg, for prompt or prefix tuning). As such, I think it'll garner interest from the NeurIPS community.

**Weaknesses:**

1. Experimental design: given that there is no theory, I find the experimental section a bit light. First, the authors only include results with OpenAI models (GPT-3 and GPT-4) which are not open-source. So it's not clear if meta-in-context learning works as well (or at all?) with publicly available LLMs. Second, can *any* LLM meta-learn in-context? How does the meta-learning ability improve with model/data size? Third, meta-in-context is only tested on small scale and toy datasets -- even the "real-world" ones are only require simple linear regression models. It'd be much more compelling if the authors could show that meta-in-context learning also improves upon in-context-learning on standard NLP tasks. NLI-type tasks would be a promising start but I'd find question answering (eg, SQuAD) or even GSM8K much more interesting.
2. Novelty: The proposed method is similar to "MetaICL: Learning to Learn In Context" by Min et al. and published at NAACL in 2022 -- yet, the authors don't even mention this work. As far as I understand, MetaICL pretrains the model such that the LLM weights are trained to adapt quickly to in-context prompts (hence they have learned to learn), while this work demonstrates that meta-learning emerges even with the standard LLM training objective. It'd be insightful to compare the two approach to quantify how much meta-training helps to adapt quickly. Min et al.'s work is especially relevant because it demonstrates benefits on real NLP tasks.
3. Scholarship: Many references in the bibliography are broken. This might seem a detail but not when a third of the references don't even state a venue.

**Questions:**

- Do open-source models also exhibit meta-in-context learning? Why or why not?
- How does meta-in-context learning depend on the size of the model / data it is trained on?
- Can meta-in-context learning outperform in-context learning on standard NLP tasks like NLI, SQuAD, or GSM8K?
- How is this work different from the Min et al.'s MetaICL paper?
- Please fix your bibliography.

**Limitations:**

The limitations of the work are appropriately discussed.

---

> ### Author Rebuttal · Authors · 2023-08-09
>
> Dear Reviewer ShPz,
>
> We appreciate that the reviewer found our paper well-motivated, clear enough to be replicated easily and the idea elegant and possibly impactful. We also thank the reviewer for their helpful comments and we have made a response to each of their comments along with suggested changes to the paper.
>
> >  First, the authors only include results with OpenAI models (GPT-3 and GPT-4) which are not open-source.  So it's not clear if meta-in-context learning works as well (or at all?) with publicly available LLMs
>
> We appreciate the feedback and agree that some analysis on other LLMs (especially open-source models which are transparent on training and architectures) does improve the empirical analysis. Therefore, we conducted additional simulations of meta-in-context learning on the latest open-source models (MPT-30b, Falcon-40b, Llama-2-7b/13b/70b). We will add the following paragraph to our revised paper:
>
> "**Meta-in-context learning in open-source models**: To comprehensively explore meta-in-context learning, we extended our evaluation to encompass five distinct open-source models. This expansion allowed us to examine whether this phenomenon is also applicable to less opaque model architectures. Notably, while the majority of models  (Falcon-40B \cite{falcon40b} and Llama-2 \cite{touvron2023llama2} models) exhibited no indications of meta-in-context learning, intriguingly, MosaicML's MPT-30B \cite{MosaicML2023Introducing} demonstrated this ability. The results are included in the Supplementary material."
>
> The figure with the corresponding results can be found in the attached PDF under Figure 6B.
>
> > How does the meta-learning ability improve with model/data size?
>
> We thank the reviewer for the feedback as we concur with the significance of investigating whether meta-in-context learning emerges as a function of model or data size. In response, we will incorporate the subsequent section into our revised paper:
>
> "**Meta-in-context learning is an emergent phenomenon**:
> In order to gain a deeper understanding of the phenomenon's characteristics, we pursued an examination into the progression of meta-in-context learning proficiency in relation to both model complexity and dataset size. To this end, we undertook an analysis encompassing smaller GPT-3 models, specifically text-ada-001, text-babbage-001, and text-curie-001 \cite{openaiAPI}. This analysis (which can be found in the Supplementary material) revealed a noteworthy trend, wherein solely text-davinci-002 seems to exhibit meta-in-context learning capabilities, thereby making it an emergent phenomenon."
>
> The figure with the corresponding results can be found in the attached PDF under Figure 6A.
>
> > It'd be much more compelling if the authors could show that meta-in-context learning also improves upon in-context-learning on standard NLP tasks.
>
> We appreciate this suggestion which was echoed by all other reviewers as well. We have therefore conducted additional simulations of meta-in-context learning on the Massive Multitask Language Understanding (MMLU) benchmark. We will add the following section to our revised paper:
>
> "**Meta-in-context learning on natural language processing benchmarks**:
>
> Finally, we examined whether meta-in-context learning also improves upon in-context-learning on standard natural language processing tasks. To test this, we conducted an experiment on the Massive Multitask Language Understanding (MMLU) benchmark \cite{hendrycks2020measuring}.
>
> **Methods**:
> We focus on the tasks from the STEM supercategory as other supercategories -- together with the addition of meta-in-context learning -- cause prompt lengths to exceed the limits of GPT-3. For the in-context learning simulations, we provided the model with $k \in {0, 1, 2}$ examples from the same category before prompting it on the test question. For the meta-in-context learning simulations, we additionally prepended three examples of two tasks from different categories.
>
> **Results**:
> Figure 9 summarizes our results. We found that meta-in-context learning was in general beneficial in terms of performance. The biggest benefit was observed in the zero-shot case, in which meta-in-context learning reached an accuracy of $55.1$ percent outperforming in-context by $22.4$ percent. This illustrates that LLMs do not necessarily have to be prompted by examples from the same category but that they can also transfer some knowledge from different categories."
>
> The figure with the corresponding results can be found in the attached PDF under Figure 9.
>
> > 2. Novelty: The proposed method is similar to "MetaICL: Learning to Learn In Context" by Min et al. and published at NAACL in 2022 -- yet, the authors don't even mention this work.
>
> Thanks a lot for mentioning this paper – we had somehow missed it. We have added it to the related work section, where we also describe how it differs from our approach:
>
> "Meta-in-context versus classical meta-learning schemes:
> [...] **Min et al. \cite{min2021metaicl} proposed a method called \emph{meta-training for in-context learning} following the same paradigm. Their work starts with a pretrained language model that is then trained (via an adjustment of its weights) on a large set of training tasks to do in-context learning, ultimately leading to a model with improved in-context learning capabilities.** In contrast to these approaches, our approach adapts a model to a series of learning problems entirely through the context itself as opposed to updating weights."
>
> > 3. Scholarship: Many references in the bibliography are broken. This might seem a detail but not when a third of the references don't even state a venue.
>
> Thanks a lot for pointing out this mistake. We strive to take good scholarship seriously and therefore went through all references to fix cases that had missing venues or other incomplete information.

---

> > ### Comment · Reviewer_ShPz · 2023-08-21
> > **Thanks for the extensive updates**
> >
> > Thank you for the many updates. Here are a few more thoughts and some smaller concerns:
> >
> > * The open-source and smaller-scale experiments look surprising — **thanks!** It's especially interesting to me that Meta-ICL works for MPT-30B but doesn't for the other open-source models, even when they are stronger (eg, LLaMa 2). Interestingly, it's also the model that gets the lowest MSE. Do you have a hypothesis for why that's the case? Could it be that the linear function learning benchmark isn't well suited for these models that are heavily text-optimized?
> >
> > * On the MMLU tasks: it looks like ICL is overtaking MetaICL already at 2 few-shot exemplars. This looks like a negative result for MetaICL on text tasks, which is still interesting. Were you able to tune the number of in-context tasks for MetaICL or was the context length a limiting factor? What accuracy does ICL reach if we allow it to use a similar context length as MetaICL? In other words, can we use MetaICL to make up for a lack of labelled exemplars in one task by leveraging other related tasks?
> >
> > * Regarding Min et al., 2022: thanks for mentioning it. Given that it's an older work, I think this submission could be further strengthened by showing if Min et al.'s pretraining objective is necessary to get the best meta-icl performance. For example, can the open-source models learn to meta-icl if pretrained / finetuned with this objective?

---

> > > ### Author Response · Authors · 2023-08-22
> > >
> > > We thank the reviewer for the feedback and tried to answer the best we can their questions due to the limit on time:
> > >
> > > - We believe that every LLM behaves differently in different benchmarks and so it is difficult to speculate about one being better in performance for a given task. Nonetheless, our meta-in-context learning experiments require very long context sizes. Therefore, we hypothesize the following which we will add to the **Meta-in-context learning in open-source models** section of our paper:
> > > *" We believe that models trained in a regime with a longer context window are more suitable to show the emergence of this phenomenon. Indeed, mpt-30b has a context-size of 8000 tokens as opposed to the LLaMa-2 models which have been trained with a context window of 4096 tokens. This is speculative and we leave the analysis of what factors influence the emergence of meta-in-context learning as a future research question."*
> > > - The price induced by the context length was the main constraint and therefore we only ran the benchmark using 3 examples per task. The similar context length condition is an interesting suggestion, and we will run the analysis for the camera-ready paper and include it in the Supplementary Material.
> > > - Finally, we agree that this line of work would be of great interest for future work but we believe it is out of scope for this project. Therefore we added the following to our **Discussion**:
> > > *"A promising direction for further research involves delving into the factors that contribute to the emergence of the meta-in-context learning phenomenon. One potential factor, as mentioned earlier, is the length of the context window. Another avenue to explore is to assess whether specialized models fine-tuned for in-context learning, like the framework proposed by Min et al. \cite{min2021metaicl} (Meta-training for In-Context Learning), yield optimal performance in the context of meta-in-context learning."*

---

### Author Rebuttal · Authors · 2023-08-09

We would like to thank all reviewers for their valuable and thoughtful feedback.

* Reviewer ShPz found “the paper an enjoyable read” and stated that “the [our] research is relatively well motivated, the analysis are carefully detailed, and the method is clearly explained.”
* Reviewer xRJY said that the “experimental setup is clean and the obtained experimental results convey a clear message.”
* Reviewer 3waD said that our paper “is well-written, and the results are thoroughly investigated and clearly conveyed.”
* Reviewer 6CRk called the paper “clearly written and easy to follow.”

Furthermore, the paper scored an average of 3.0 on both soundness and presentation.

However, all reviewers also raised important points and provided helpful suggestions. We were able to incorporate all of these suggestions and believe that doing so has improved our paper significantly. To summarize, we have made the following modifications:

* We tested meta-in-context learning on a natural language processing benchmark (MMLU), where we found good performance (requested by all reviewers).
* We have included experiments with **eight** additional models to investigate:
 	1. whether meta-in-context learning is an emergent phenomenon that arises at scale (reviewer ShPz).
 	2. whether meta-in-context learning can also be found in open-source models (reviewer ShPz).
* We ran additional experiments with non-linear (quadratic functions), where we also found meta-in-context learning to be beneficial (reviewer 6CRk).
 * We performed additional analyses to investigate how task similarity affects meta-in-context learning performance (reviewers xRJY).
 * We incorporated references suggested by the reviewers (reviewers NP3F, xRJY, ShPz).
 * We added more details on the probit regression analysis from the two-armed bandit task (reviewers 3waD).
* We fixed reference issues, typos and improved the structure slightly.

We describe these changes in detail in our responses to the individual reviews below. We again want to thank the reviewers for their time and for actively taking part in the review process.

---

### Comment · Area_Chair_jAuM · 2023-08-10
**Author-Reviewer Discussion phase (Aug 10-16)**

Today begins the Author-Reviewer Discussion phase, which lasts 1 week (**Aug 10-16**).

I ask the reviewers to please **carefully read all other reviews and the author responses
and (if appropriate) respond to author responses promptly.**   If you've read the author response, please take the time to leave a comment, even if you have nothing to add.

I also encourage both authors and reviewers to monitor OpenReview for further comments in order to enable as much back-and-forth as possible during this short period.

The authors response mentions a number of new experimental results which may address some reviewers' concerns, and I also note concerns about the practical implications which may deserve further discussion during this phase.

---

### Decision · Program_Chairs · 2023-09-21

**Decision:**

Accept (poster)

**Comment:**

This paper documents meta-in-context learning, whereby including a set of tasks in a prompt can provide additional improvements compared with only examples from the intended task.  Personally, I don’t find this phenomenon surprising in light of in-context learning, but I agree that it is worth identifying and studying, and will be relevant to the NeurIPS community.  Reviewers seem satisfied with the experiments, presentation, and overall quality of the work, especially after the authors’ response and updates.  Furthermore, as highlighted in discussions with reviewers, meta-in-context learning could prove useful for practitioners in small data settings.  I recommend accepting the paper, and expect that the authors will ensure their updated manuscript includes all of the necessary changes.